# Assessment of Some Risk Factors and Biological Predictors in the Post COVID-19 Syndrome in Asthmatic Patients

**DOI:** 10.3390/jpm14010021

**Published:** 2023-12-22

**Authors:** Oana Elena Melinte, Daniela Robu Popa, Mona Elisabeta Dobrin, Andrei Tudor Cernomaz, Cristina Grigorescu, Alexandra Floriana Nemes, Adina Catinca Gradinaru, Cristina Vicol, Doina Adina Todea, Damiana Maria Vulturar, Ionel Bogdan Cioroiu, Antigona Carmen Trofor

**Affiliations:** 1Discipline of Pneumology, III-rd Medical Department, Faculty of Medicine, “Grigore T. Popa” University of Medicine and Pharmacy, 700115 Iasi, Romania; oana-elena.melinte@umfiasi.ro (O.E.M.); tudor.cernomaz@umfiasi.ro (A.T.C.); cristina.vicol@umfiasi.ro (C.V.); antigona.trofor@umfiasi.ro (A.C.T.); 2Clinical Hospital of Pulmonary Diseases, 700116 Iasi, Romania; elisabetamona.dobrin@pneumo-iasi.ro; 3Discipline of the Thoracic Surgery, Faculty of Medicine, “Grigore T. Popa” University of Medicine and Pharmacy, 700115 Iasi, Romania; cristina.grigorescu@umfiasi.ro; 4Doctoral School, Faculty of Medicine, “Titu Maiorescu” University, 031593 Bucharest, Romania; alexandra.nemes@prof.utm.ro; 5Discipline of Pharmaceutical Botany, Faculty of Pharmacy, “Grigore T. Popa” University of Medicine and Pharmacy, 700115 Iasi, Romania; adina.gradinaru@umfiasi.ro; 6Discipline of Pneumology, Department of Medical Sciences, Faculty of Medicine, “Iuliu Hatieganu” University of Medicine and Pharmacy, 400012 Cluj, Romania; dtodea@umfcluj.ro (D.A.T.); vulturar.damianamaria@elearn.umfcluj.ro (D.M.V.); 7Romanian Academy-Iasi Branch, Research Center for Oenology, 700490 Iasi, Romania; bogdan.cioroiu@acadiasi.ro

**Keywords:** post-COVID-19 syndrome, asthma, comorbidities, biological variables

## Abstract

Long COVID-19 or post-COVID infection (PCI) refers to the prolongation of symptoms in people who have been infected with the SARS-CoV-2 virus. Some meta-analysis studies have shown that patients with comorbidities, such as diabetes, obesity or hypertension, have severe complications after infection with the SARS-CoV-2 virus. The presence of chronic respiratory diseases such as bronchial asthma, COPD, pulmonary hypertension or cystic fibrosis increases the risk of developing severe forms of the COVID-19 disease. The risk of developing the severe form of COVID-19 was observed in patients with bronchial asthma being treated with corticosteroids, but also in those hospitalized with severe asthma. The biological variables determined in patients with PCI infection showed changes, especially in the hematological parameters, but also in some inflammatory markers. The aim of this study was to investigate some biological predictors in post-COVID-19 infection in patients with asthma and various comorbidities. In the case of patients diagnosed with moderate and severe forms of COVID-19, the variation in biological tests has shown high concentrations for serum glucose, lactate dehydrogenase and C-reactive protein. Additionally, the calculation of the relative risk (RR) based on the associated comorbidities in patients with PCI points to higher values for patients with asthma, hypertension, diabetes and obesity (RR moderate/severe form = 0.98/1.52), compared to patients with PCI and asthma (RR moderate/severe form = 0.36/0.63). Based on the statistical results, it can be concluded that the alanine aminotransferase (ALT) activity (*p* = 0.006) and the age of patients (*p* = 0.001) are the variables that contribute the most to the separation of the four classes of comorbidities considered.

## 1. Introduction

The National Institute for Health and Care Excellence (NICE) has defined “post-COVID-19” syndrome as the “signs and symptoms that develop during/after a SARS-CoV-2 infection which persist for more than 12 weeks and cannot be explained by any other diagnosis” [1]. It has been observed that these persistent symptoms may be due to the presence of inflammation, prolonged ventilation, social isolation, non-specific effects of hospitalization, and the presence of comorbidities [2,3]. The risk of reinfection is a major topic of debate regarding the COVID-19 pandemic. Many studies have established a connection between the presence of comorbidities and the incidence of post-COVID-19 infection. Furthermore, recent studies reported that the presence of comorbidities is associated with the severity of the COVID-19 infection; the most common comorbidities include diabetes, cardiovascular diseases, hypertension, cancer, oxidative stress, kidney disorders, and vitamin deficiencies [4,5]. Coexisting respiratory diseases such as asthma, chronic obstructive pulmonary disease (COPD), pulmonary hypertension, or cystic fibrosis increase the risks of severe COVID-19 [6]. According to the recent update from the Global Initiative for Asthma (GINA), asthma is a heterogeneous disease typically characterized by chronic airway inflammation. Asthma is defined as a chronic disease characterized by symptoms such as wheezing, shortness of breath, dyspnea and cough, along with reduced expiratory airflow [7]. A recent update of the GINA and COVID-19 guidelines concluded that, based on current evidence, individuals with well-controlled mild to moderate bronchial asthma do not have an increased risk of developing COVID-19. They also have a lower risk of developing severe forms of COVID-19 [8]. The risk of developing severe COVID-19 or the risk of death from SARS-CoV-2 infection has been observed, especially in individuals with asthma who are on oral corticosteroid treatment and those hospitalized for severe asthma [8]. Studies have also shown that patients with asthma have an increased risk of exacerbations after being infected with respiratory viruses, such as the flu virus, parainfluenza virus and SARS-CoV-2 [9].

It was observed that patients who had a severe form of COVID-19 or experienced complications during the infection required monitoring even after testing negative for SARS-CoV-2 by RT-PCR. Evaluating the specific biomarkers for heart, kidney, liver, lung, diabetes, inflammation, and other infections is mandatory if the patient experiences persistent symptoms at 12 weeks or even 6 months after the infection [10]. Biomarkers associated with post-COVID-19 infectious syndrome may include elevated C-reactive protein (CRP) levels and serum ferritin, which are often correlated with disease severity and prognosis [11]. Meta-analysis studies have shown that elevated CRP levels are directly related to inflammation and disease severity, indicating the development of lung lesions in the early stages of the disease [12]. Therefore, CRP is considered the most effective and sensitive biomarker for predicting the progression of COVID-19. It has been observed that coagulation tests are usually within normal limits, but D-dimer values are elevated [13]. Hematological values in patients with post-COVID-19 infectious syndrome are typically characterized by changes in the number of lymphocytes, neutrophils, leukocytes, and platelets. Leukocytosis, especially neutrophilia, represents another modification induced by SARS-CoV-2 infection and has been observed in a large number of patients diagnosed with COVID-19 [14,15,16]. Lymphopenia is common and occurs in approximately 80% of patients. Thrombocytopenia (a low platelet count) is usually mild (platelets are rarely < 100). In severe cases, severe thrombocytopenia can be a poor prognostic sign of the disease [17].

This study aims to evaluate the epidemiological and clinical risk factors associated with developing post-COVID-19 infection (PCI) among patients with asthma. Therefore, the main risk factors evaluated were the clinical comorbidities associated with patients diagnosed with asthma during the pandemic who subsequently became reinfected and were diagnosed with PCI.

## 2. Materials and Methods

In the present study, a group of 37 Caucasian patients diagnosed with bronchial asthma and multiple comorbidities were selected, and they experienced later a reinfection with the SARS-CoV-2 virus. This work represents a retrospective observational study that includes patients with asthma who were diagnosed with post-COVID-19 infection (PCI) approximately 3–6 months after their initial infection.

The patients included in this research were investigated throughout their hospitalization at the Clinical Hospital of Pulmonary Diseases in Iasi, Romania; a hospital serving as a regional reference center for respiratory disorders in the North-East (Moldova) region of the country. We identified patients with asthma developing moderate and severe forms of COVID-19 in 2020 and who later who experienced reinfection with the SARS-CoV-2 virus within a period of 30–180 days.

The study included patients diagnosed with bronchial asthma (BA) and other comorbidities, including hypertension (H), diabetes mellitus (DM), obesity (Ob), chronic renal failure, and prostate adenoma. Exclusion criteria were: patients under the age of 18, patients diagnosed with chronic obstructive pulmonary disease (COPD) or cystic fibrosis, and pregnant women.

The clinical and demographic characteristics of the investigated patients are presented in Table 1. The demographic data refer to age, gender, residential area (urban/rural), and smoking history (Table 1). Laboratory investigations were extracted from the hospital’s electronic system. Laboratory assessments included biochemical parameters (ALT-alanine aminotransferase, AST-aspartate aminotransferase, urea, creatinine, glucose, lactate dehydrogenase (LDH), C-reactive protein (CRP)), and complete blood counts (CBC). The laboratory markers were monitored based on comorbidities in patients with asthma who had a history of moderate/severe COVID-19 and were diagnosed with post-COVID-19 infection 30–180 days after their initial infection.

### 2.1. The Determination of Biochemical Parameters

The quantification of biochemical parameters was carried out using the automated analyzer Cobas Integra 400 plus (Roche Diagnostic, Indianapolis, IN, USA). All tests were examined in the biochemistry laboratory, and quality assurance procedures, including internal quality control and external quality control, complied with the rules.

### 2.2. The Determination of Hematological Parameters

The quantification of hematological parameters in patients diagnosed with COVID-19 was performed using flow cytometry with the automated analyzer from Sysmex (Sysmex Corporation, Japan). A complete blood count (CBC) was conducted, and the following hematological parameters were evaluated in the present study: white blood cells (WBC), red blood cells (RBC), hemoglobin (HGB), hematocrit (HCT), platelets (PLT), lymphocytes (L), monocytes (M), eosinophils (EO) and neutrophils (N).

### 2.3. The Statistical Analysis

Descriptive statistics for paraclinical variables are expressed in terms of means, standard deviations, minimum, and maximum values. Statistical differences between the investigated groups, patients with moderate COVID-19 versus patients with severe COVID-19, were determined using the non-parametric Mann–Whitney U test. Non-parametric tests (Spearman) were used to identify possible correlations between the biochemical and hematological markers of the subjects under investigation. To evaluate the risk factors associated with disease severity, multivariable logistic regression analysis was applied using Statistica10. The relative risk (RR) was calculated based on the investigated comorbidities. Discriminant analysis was used to monitor the profile of the biochemical parameters in SARS-CoV-2 infection, with the goal of differentiating patients with post-COVID-19 infection and various associated comorbidities based on the investigated biological variables.

### 2.4. Ethics Consideration

The study was approved by the ethics committee of the Clinical Hospital of Pulmonary Diseases, Iasi, Romania (ethical approval No. 96/16.03.2023) and by the ethics committee of the University of Medicine and Pharmacy “Grigore T. Popa” Iasi, Romania (ethical approval No. 319/30.05.2023).

## 3. Results

A total of 37 patients diagnosed with asthma and multiple comorbidities were investigated at the Clinical Hospital of Pulmonary Diseases Iasi between 2020 and 2021. After the initial infection with the SARS-CoV-2 virus, the patients returned to the medical institution between 30 and 180 days later, when they were examined and diagnosed with post-COVID-19 syndrome. The demographic and clinical data of the investigated patients are presented in Table 1.

Patients with asthma and severe COVID-19 presented multiple comorbidities, such as hypertension (n = 12), diabetes (n = 6), and obesity (n = 4). They also had a longer duration of hospitalization, approximately 7.26 ± 5.85 days (Table 1). In the case of moderate COVID-19, there were 19% male and 81% female patients. The severe COVID-19 form was represented by 63% male and 37% female patients. Numerous studies have investigated gender as a risk factor in post-COVID-19 infection and have observed that older women with higher body weight are more susceptible to infection than men [18].

Among patients with asthma who developed a moderate form of COVID-19 during the pandemic, there was a prevalence of hypertension (n = 14), diabetes (n = 3), and obesity (n = 3). Other comorbidities present among the investigated patients included chronic kidney disease and chronic liver disease.

To control asthma symptoms and improve their breathing capacity, patients diagnosed with post-COVID-19 infection were recommended oral corticosteroids by the physician. The corticosteroids used included beclomethasone, fluticasone, budesonide, and mometasone, administered in doses corresponding to the severity stage of asthma. Upon admission, significant changes in laboratory analyses were observed, particularly in the case of CRP, LDH, white blood cell count, neutrophil count, and lymphocyte count (Table 2).

The results of the descriptive statistics for the biochemical parameters in patients diagnosed with COVID-19 revealed values above the reference range for AST and ALT, in the cases of both moderate and severe forms. For the severe form, the average concentrations of ALT were found to be 44.43 U/L, with some patients even reaching values as high as 281.7 U/L (Table 2). The results of the statistical analysis generated for non-parametric tests indicated that the investigated data were not normally distributed (*p* < 0.05), and the Spearman correlation coefficients showed statistical significance (r = 0.4–0.89) between the investigated clinical parameters and the associated pathologies (Figure 1 and Figure 2).

The distribution of the values for the investigated clinical variables was represented using Quantile–Quantile (Q-Q) plots. Q-Q plots generated in Statistica10 show the dispersion of observed values compared to expected values, and significant variations were observed, especially for CRP, LDH, serum glucose, white blood cells, ALT, and AST (Appendix A).

Serum creatinine showed higher values, especially in patients with severe COVID-19, with maximum values recorded at 1.55 mg/dL. In the case of patients diagnosed with moderate and severe forms of COVID-19, the variation in biological tests showed high concentrations for serum glucose, LDH, and CRP. The variability in the concentrations of the biochemical parameters in the case of moderate and severe forms of COVID-19 in the patients included in the study is presented in Table 2.

In the present study, the statistical significance of the difference between patients with severe and moderate COVID-19 was evaluated using the Mann–Whitney U test. Significant statistical differences were observed in the following biochemical parameters: LDH (*p* = 0.007), CRP (*p* = 0.005), and the concentration of corticosteroids administered to patients (*p* = 0.011) (Table 3). Analyzing the serum LDH concentrations based on associated comorbidities, high values were observed in patients with asthma (average concentration of 276.64 U/L) and in patients with asthma and hypertension (average concentration of 265.85 U/L) (Table 4).

The CRP values differentiated between patients with asthma (average concentration of 59.66 mg/L) and those with asthma, hypertension, and obesity (43.66 mg/L) are described in Table 4. Additionally, the calculation of the relative risk (RR) based on associated comorbidities in patients with PCI highlighted higher values for patients with asthma, hypertension, diabetes, and obesity (RR moderate/severe form = 0.98/1.52) compared to patients with PCI and asthma (RR moderate/severe form = 0.36/0.63). Recent studies have shown that patients with multiple comorbidities (diabetes, obesity, hypertension, chronic kidney diseases) develop various clinical manifestations in PCI infection [4].

Since the beginning of the COVID-19 pandemic, an increasing number of studies have focused on the relationship between laboratory parameters and the severity of diagnosis in SARS-CoV-2 infection [19]. Thus, the goal has been to identify the biochemical and hematological variables that differentiate mild forms of COVID-19 from moderate and severe forms. Table 5 presents the results regarding the selection of the most important statistical variables. These results show that of the 17 parameters analyzed, only 5 were retained as representative for the model, meaning they significantly contribute to the separation of the proposed study classes, specifically patients with BA, patients with AB, H, DM, patients with AB, H, patients with AB, H, OB.

The statistical results for the classification functions (Root 1 and Root 2) corresponding to the four considered classes are presented below. The standardized coefficients for each variable retained in the model and for each discriminant function are shown in Table 5. Figure 3 contains the classes and their relationships by representing the individual scores of the investigated subjects along the main discriminant functions.

Risk factors, especially comorbidities, have been extensively investigated in SARS-CoV-2 infection, and it has been observed that 37% of subjects diagnosed with PCI have diabetes mellitus. Therefore, the blood glucose values were evaluated in patients with asthma and PCI. Descriptive statistics results highlighted average serum glucose concentration values of 140.97 mg/dL for patients with post-COVID-19 infection and comorbidities (asthma, hypertension, diabetes, obesity), 139.7 mg/dL for patients with asthma, hypertension, and obesity, and 128.7 mg/dL for patients with asthma, hypertension, and diabetes (Table 4, Figure 4a). The linear regression model applied to the dataset shows a positive correlation of blood serum glucose and the number of hospitalization days (*p* < 0.05, r = 0.42) in patients with asthma and multiple comorbidities Figure 4b.

Numerous tests and statistical interpretations were conducted to assess the risk of patients with diabetes and severe COVID-19. Survival data are often represented graphically through survival curves, with the Kaplan–Meier method being the most commonly used. In the current study, a survival curve was created for patients with asthma and PCI, as well as those with multiple comorbidities (hypertension, diabetes, obesity), taking into account the hospitalization period. Figure 5 represents the survival curve for patients diagnosed with PCI, but with a history of asthma and multiple comorbidities.

## 4. Discussion

The patients included in the study had an age range of 28–85 years and came from both urban (19) and rural (18) areas in the Iasi County, as well as from various locations in the Moldova region. The patients were diagnosed with the SARS-CoV-2 infection based on PCR testing, chest X-ray, and laboratory tests. This study primarily aims to evaluate the epidemiological and clinical risk factors associated with a higher risk of developing post-COVID-19 infection (PCI). We evaluated numerous comorbidities as risk factors for asthma in moderate and severe forms of the COVID-19 infection.

The severe form of COVID-19 is distinguished by elevated levels of serum LDH, and 81% of these subjects had values exceeding 225 U/L. Although LDH is an enzyme with origins in many organs and systems, its values significantly increase, especially in patients with lung disease [20,21]. There is ample evidence suggesting that elevated serum LDH reflects the extent of various pathophysiological processes [20]. LDH can be released during tissue damage and is involved in various pathophysiological processes, serving as a non-specific indicator of cell death. Several previous studies have shown that elevated serum LDH is associated with poor prognosis in malignancies [20]. In this study, serum LDH is validated for its potential utility as a marker for assessing clinical severity in SARS-CoV-2 infection. It is known that various disorders can increase serum LDH concentration, such as heart failure, hypothyroidism, hemolytic anemia, and cancer [22]. Among the subjects with asthma who were investigated, 21 were diagnosed with moderate COVID-19, and 16 with severe COVID-19. The clinical investigation presented in Table 2 showed biochemical parameter modifications, especially for CRP (mean concentration = 21.3 mg/L) and LDH (mean concentration = 215.09 U/L) in subjects diagnosed with moderate COVID-19. Studies in the literature have associated mild COVID-19 pneumonia with LDH serum concentrations < 220 U/L and CRP concentrations < 22 mg/L, while severe cases were characterized by CRP values > 80 mg/L and LDH serum values > 380 U/L [22]. An increasing number of studies have investigated obesity as a risk factor in post-COVID-19 infection. Recent studies have shown that obesity is significantly associated with post-COVID-19 infection (OR, 1.15); however, this significant correlation may not be present in all studies [18]. In this study, complete blood count (CBC) values were characterized by changes in the number of lymphocytes, neutrophils, leukocytes, and platelets (Table 2). It was observed that leukocytosis and, in particular, neutrophilia, represents another alteration induced by SARS-CoV-2 infection and were noted in a large number of patients diagnosed with COVID-19 [23,24,25]. The increase in the number of neutrophils, accompanied by a decrease in the number of lymphocytes, has been associated with the risk of death in patients diagnosed with PCI. In this study, high neutrophil counts were particularly observed in severe cases of COVID-19 (Table 2). High neutrophil counts were evaluated based on the comorbidities of the patients investigated. Thus, patients with PCI and associated comorbidities (asthma, hypertension, obesity) had an average value of N = 71.2%. For patients with asthma and PCI, the mean value for N was 65.5%, while for patients with asthma, hypertension, and PCI, N = 67.98% (Table 3). Studies in the literature have associated these increases in neutrophil counts with septic conditions and advanced organ dysfunction [26].

Numerous hypotheses have been proposed to understand the pathogenesis of lymphopenia in the context of SARS-CoV-2 infection. It seems that viral particles spread in the respiratory mucosa using the ACE2 receptor induce a cytokine storm, generating a series of immune responses, leading to changes in the number of lymphocytes, particularly CD4 lymphocytes, which have been considered biological predictors of severity and mortality. Thus, regarding the COVID-19 pathology, it has been postulated that patient survival may depend on their ability to restore the lymphocytes destroyed by the virus [27]. In terms of the eosinophil count in patients with PCI and bronchial asthma, a slight difference has been observed associated with patients with severe COVID-19 (3.31 × 10^3^/µL) compared to patients in the moderate form group (3.17 × 10^3^/µL) (Table 2). Recent studies have reported an increase in the eosinophil count in patients with bronchial asthma and COVID-19 of 5.3 ± 5.5 compared to asthmatic patients without COVID-19 of 0.735 ± 0.8570 [28]. The eosinophils play a role in the immune system and are often associated with allergic responses and parasitic infections. In the context of COVID-19, certain lower levels of eosinophils have been observed in severe cases, and an increase in eosinophils has been noted in specific situations, including in patients with bronchial asthma. In general, eosinophils are known for their role in modulating immune responses, and it has been suggested that they play a protective role against certain viral infections. However, the relationship between eosinophils and COVID-19 is complex, and the specific implications of increased eosinophils in COVID-19 patients with bronchial asthma are not fully understood. It has been suggested that eosinophils have antiviral properties, and their increased presence might be associated with a protective response against viral infections [29].

In this study, discriminant analysis aimed to differentiate patients with PCI and other associated pathologies based on clinical and paraclinical parameters. The statistical data resulting from the discriminant analysis showed that the five variables retained in the “model” (characterized by the type and number of classes) contributed most significantly to the separation of the four types of cases considered, namely patients with asthma and hypertension, patients with asthma, hypertension and diabetes, patients with asthma, hypertension and obesity and the last group of patients with asthma, hypertension, diabetes and obesity.

It should be observed that the most significant parameter in terms of contribution to the separation of the classes is serum ALT (λ-partial = 0.309, F(4.28) = 4.502, *p* = 0.006), while CRP concentrations contributed the least among the variables retained in the model (λ-partial = 0.807, F(4.28) = 1.668, *p* = 0.027). Throughout the three discriminant functions, Root 1/Root 2/Root 3, the classes showed moderate separation, with differentiation observed within the classes (Figure 3) The first discriminant function (Root 1), characterized by the highest absolute standardized coefficients (ALT, age of the investigated patients), achieved good separation of patients with PCI and AB from those with AB, H, DM, patients with AB, H, patients with AB, H, OB. Discriminant analysis applied to the dataset presented the profile of laboratory parameters (independent variable) according to the diagnosis of patients (grouping variable) in the case of patients with moderate/severe COVID-19. Based on the statistical results, it can be concluded that ALT activity (*p* = 0.006) and the age of patients (*p* = 0.001) are the variables that contribute the most to the separation of the four classes considered. Most studies have observed that ALT values increase transiently in COVID-19, due to liver dysfunction caused by the systemic inflammatory response following the administration of hepatotoxic drugs [30]. Thus, the most statistically significant correlations were observed in patients with AB-H-DM-Ob for ALT parameters and the number of days of hospitalization (r = 0.54, *p* < 0.05), patients with AB-H-DM for urea serum and hospitalization days (r = 0.88; *p* < 0.05), but also for serum glucose and neutrophil counts in AB patients (r = 0.63, *p* < 0.05), and the number of neutrophils with the values of CRP concentrations in patients with AB-H-Ob (r = −0.988, *p* < 0.05) (Figure 1 and Figure 2).

Numerous studies have considered age as a risk factor in post-COVID-19 infection, observing that older women with high body weight (obesity) are much more susceptible to the infection than men [18]. Meta-analyses have outlined the profile of patients susceptible to post-COVID-19 infection, and it has been observed that age, female sex, severe form of COVID-19, the presence of at least five symptoms during the acute phase of the disease, and obesity are the most representative characteristics of this condition [31]. Not only obesity but also diabetes are the main risk factors that increase mortality in patients with COVID-19. From the available data, it can be seen that the prognosis of patients with diabetes is significantly influenced by the SARS-CoV-2 infection. Recent studies have shown a prevalence for diabetes of 5.7–5.9% in patients with moderate COVID-19 and of 22.2–26.9% in severe COVID-19 [24]. Numerous tests and statistical interpretations have been performed to assess the risk in patients with diabetes and severe COVID-19. Along with other chronic disorders, it has been observed that diabetes increases the risk of complications in severe acute respiratory distress syndrome (ARDS) [32]. Moreover, in this study, the clinical investigation in patients with diabetes and severe COVID-19 presented higher serum concentrations of CRP and LDH. Other studies have demonstrated a direct association between CRP and LDH concentrations and the severity of the diagnosis [33,34]. These results indicated that patients with diabetes diagnosed with severe COVID-19 produced a more severe inflammatory response compared to patients diagnosed with a moderate form of COVID-19. The incidence of high LDH values in patients with COVID-19 has been associated with the presence of diabetes. This phenomenon can be explained by the reduction in glycogen synthesis and changes in glucose oxidative metabolism [35]. This observation may suggest that patients with diabetes had a more severe form of COVID-19 due to pancreatic damage caused by the virus and, in particular, the cytokine storm [24,36].

The results of the present study indicated high serum glucose concentrations in patients with severe forms of COVID-19 and multiple comorbidities, but also a statistical correlation between hospitalization days and serum glucose concentration (r = 0.42 *p* < 0.05 (Figure 4a,b) The literature studies showed that patients with COVID-19 and diabetes developed a severe inflammatory response, needed mechanical ventilation and at the same time had a greater number of days of intensive care hospitalization [24].

It has also been stated that a deficiency of vitamin D is one of the causes of increased plasma glucose and dyslipidemia. Thus, in a study conducted on children with asthma in North America, it was observed that low levels of vitamin D were directly associated with an increased risk of hospitalization, resulting in a significant inverse relationship between serum vitamin D levels and the severity of asthma [37]. Other research studies hav e shown that the severity of COVID-19 was significantly different in patients based on their serum calcium levels; the lower the calcium level, the more severe the disease [38].

The Kaplan–Meier survival curve for patients with PCI and multiple comorbidities (Figure 5) does not show significant differences between patients with asthma and those with asthma/hypertension/diabetes/obesity, but better results are seen in patients who only have the diagnosis of asthma. A study with a larger number of patients with asthma versus patients with multiple comorbidities could potentially identify more significant differences in this area of research.

## 5. Conclusions

This study represents an analysis of the clinical and paraclinical characteristics of patients diagnosed with post-COVID-19 infection, focusing on the variability of laboratory parameters based on the associated comorbidities. It was observed that patients with post-COVID-19 infection exhibited elevated inflammatory biochemical markers after the initial infection. The values of CRP, LDH, and the concentration of corticosteroids administered to the investigated patients highlighted a statistical difference between the severe and moderate forms of COVID-19. Furthermore, it was observed that certain characteristics such as age and comorbidities were associated with an increased risk of post-COVID-19 infection. The relative risk (RR) calculated based on the associated morbidity was higher in patients with post-COVID-19 infection and multiple comorbidities (asthma, hypertension, diabetes, obesity) compared to patients who only had asthma. Consequently, it can be concluded that bronchial asthma is not a risk factor for an unfavorable prognosis in COVID-19. Yet, older age and coexisting multiple comorbidities in asthmatic patients should be an alert for an increased effort to maintain asthma control and to prevent the risk of acute exacerbations that may precipitate disease progression to respiratory failure or death and at the same time could increase COVID-19-related mortality.

## 6. Limitations

The limitation of this study could be the low addressability of asthmatic patients who were diagnosed with post-COVID-19 syndrome, at a distance from the acute viral episode, reason for which the group of patients included in this study was limited, this fact being emphasized in what concerns corticotherapy as a risk factor in the development of severe forms of the disease.

## Figures and Tables

**Figure 1 jpm-14-00021-f001:**
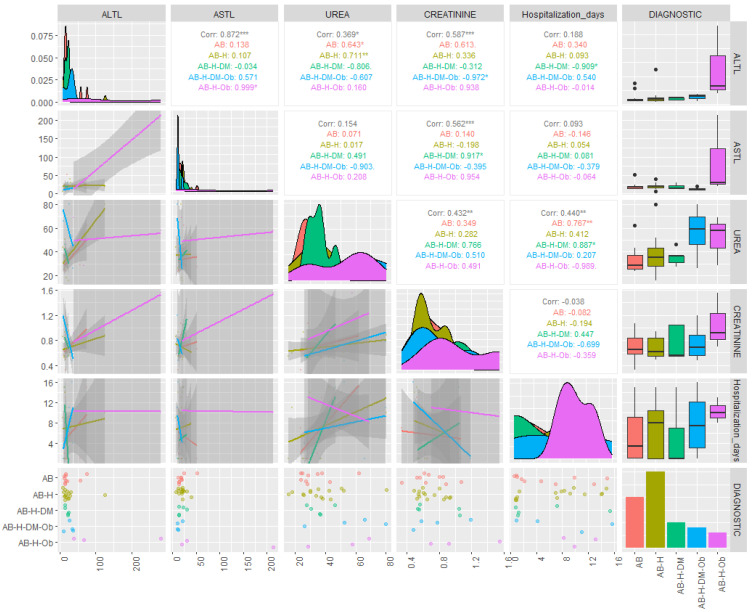
Correlations of biochemical parameters based on associated comorbidities in patients with asthma and PCI. (ALTL = TGP: alanine aminotransferase; ASTL = TGO: aspartate aminotransferase; AB = bronchial asthma; AB-H= bronchial asthma-hypertension; AB-H-DM = bronchial asthma-hypertension-diabetes mellitus; A-H-DM-Ob= bronchial asthma-hypertension-diabetes mellitus-Obesity; AB-H-Ob = bronchial asthma-hypertension-obesity; corr = correlation; * *p* ˂ 0.05 ** *p* ˂ 0.005; *** *p* ˂ 0.001).

**Figure 2 jpm-14-00021-f002:**
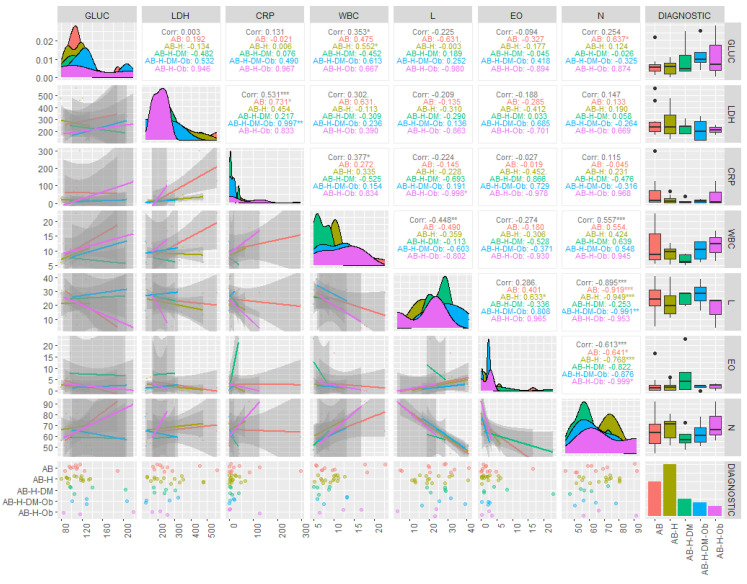
Correlations of biochemical and hematological parameters based on associated comorbidities in patients with asthma and PCI. (Gluc = glucose (mg/dL); LDH = lactat dehydrogenase (U/L); CRP = C-reactive protein (mg/L); WBC = white blood count (×10^3^/µL); L = lymphocites (×10^3^/µL); EO = Eosinophils (×10^3^/µL); N = Neutrophils (×10^3^/µL), AB = bronchial asthma; AB-H = bronchial asthma-hypertension; AB-H-DM = bronchial asthma-hypertension-diabetes mellitus; AB-H-DM-Ob = bronchial asthma-hypertension-diabetes mellitus-Obesity; AB-H-Ob = bronchial asthma-hypertension-Obesity; corr = corelation; * *p* ˂ 0.05 ** *p* ˂ 0.005; *** *p* ˂ 0.001).

**Figure 3 jpm-14-00021-f003:**
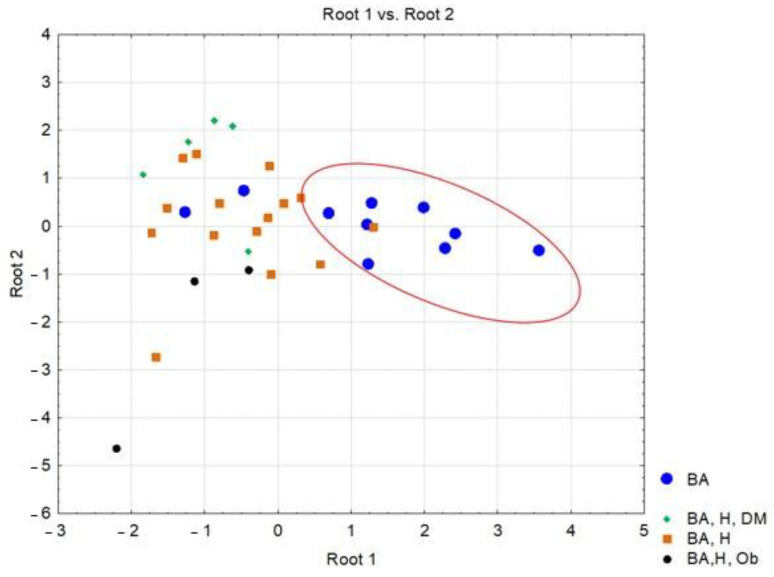
Representation of scores along the two discriminant functions corresponding to patients diagnosed with post-COVID-19 infection, based on associated comorbidities.(BA = bronchial asthma; BA, H, DM = bronchial asthma, hypertension-diabetes mellitus; BA, H = bronchial asthma, hypertension; BA, H, Ob = bronchial asthma, hypertension, Obesity).

**Figure 4 jpm-14-00021-f004:**
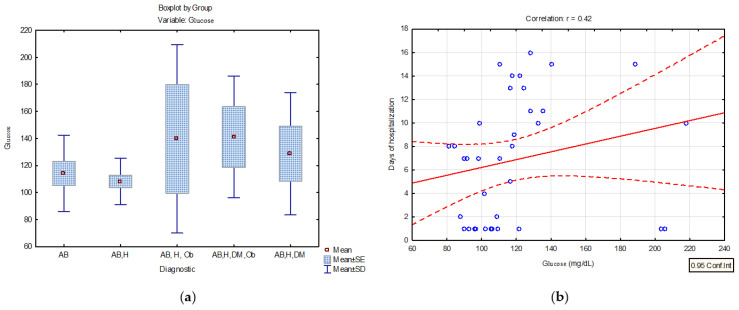
(**a**) Distribution of serum glucose concentrations (mg/dL) in patients diagnosed with PCI and various associated comorbidities (BA = bronchial asthma; AB, H = bronchial asthma, hypertension; AB, H, Ob = bronchial asthma, hypertension, obesity AB, H, DM, Ob = bronchial asthma, hypertension, diabetes mellitus, obesity; AB, H, DM- bronchial asthma, hypertension, diabetes mellitus) (**b**) linear regression curve between serum glucose concentration and the number of days of hospitalization in patients with PCI and different comorbidities.

**Figure 5 jpm-14-00021-f005:**
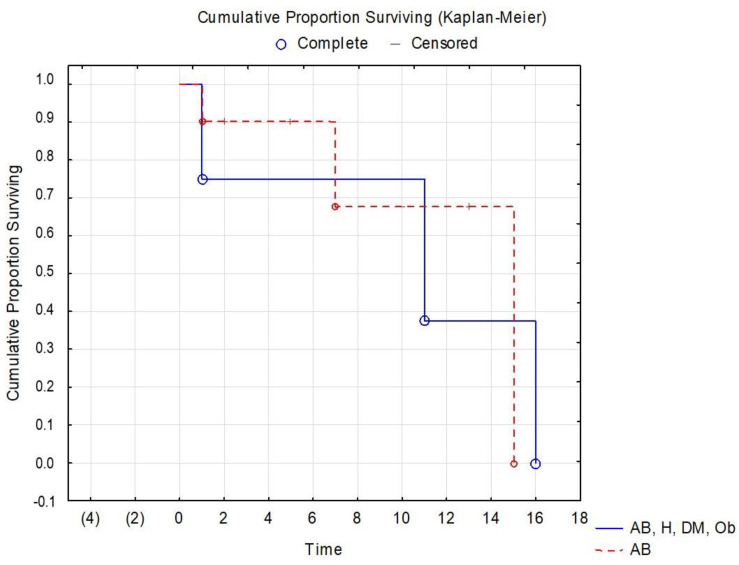
Kaplan–Meier survival curve for patients with PCI and various comorbidities.(AB = bronchial asthma, AB, H, DM, Ob = bronchial asthma, hypertension, diabetes mellitus, Obesity).

**Table 1 jpm-14-00021-t001:** Demographic and clinical characteristics of post-COVID-asthma patients.

Characteristics of the Patients	Moderate Form	Severe Form
	COVID-19	COVID-19
	(n = 21)	(n = 16)
**Variable**	Mean ± SD	Mean ± SD
Age (years)	66.71 ± 14.04	61.61 ± 14.31
Gender		
Male (n/%)	4 (19%)	10 (63%)
Female (n/%)	17 (81%)	6 (37%)
Residential area		
Rural area (n/%)	9 (43%)	9 (56%)
Urban area (n/%) Smoking status	12 (57%)	7 (44%)
Ex-smokers/non-smokers	18 (49%)/4 (11%)	8 (22%)/7 (18%)
Hospitalization days	5.43 ± 4.9	7.26 ± 5.85
**Comorbidities**		
Hypertension (n)	14 (66%)	12 (75%)
Diabetes mellitus (n)	3 (14%)	6 (37%)
Obesity (n)	3 (14%)	4 (25%)
Chronic kidney diseases n)	1 (5%)	1(6%)
Chronic liver diseases (n)	1(5%)	1(6%)
Number of deaths (n)	1(5%)	4 (25%)
**Treatment with corticosteroids**		
Beclomethasone (mcg/day) n/%	(400–1000) 11/55%	(200–1000) 6/46%
Fluticasone (mcg/day) n/%	(200–500) 3/15%	(1000) 3/23%
Budesonide (mcg/day) n/%	(320–640) 4/20%	(320–1200) 4/31%
Mometasone (mcg/day) n/%	(62.5–136) 2/10%	-

**Table 2 jpm-14-00021-t002:** Descriptive statistics (mean, standard deviation, minimum and maximum values) for patients with asthma and post-COVID-19 infection.

		Moderate Form			Severe Form	
	Mean	Stdev	Range	Mean	Stdev	Range
ALT	25.90	26.30	8.2–126	44.43	64.02	8.20–281.7
AST	19.09	8.96	6.9–51.70	30.79	45.90	6.9–214.40
Urea	35.79	15.47	15.9–80.30	42.76	19.77	15.4–80.30
Creatinine	0.67	0.15	0.49–0.94	0.78	0.33	0.15–1.55
Glucose	111.11	25.92	84.6–203.20	124.84	48.54	25.9–218.0
LDH	215.09	74.38	133–456	290.48	133.62	74.38–560
CRP	21.93	64.07	0.6–298	52.52	71.09	0.60–298.2
WBC	8.24	2.19	5.14–12.66	10.70	5.37	2.19–22.69
RBC	4.47	0.57	3.23–5.60	4.42	1.23	0.57–6.02
HGB	13.03	1.78	9.9–16.30	12.74	3.49	1.78–17.30
HCT	39.21	5.20	29.4–49.40	38.45	10.36	5.20–52.0
PLT	282.71	76.51	122–434	268.34	121.81	76.51–491
L	25.25	8.07	10.9–41	19.70	11.21	3.50–41.10
M	8.35	3.19	4.7–17.6	7.95	3.88	2.00–17.60
EO	3.17	3.73	0.1–16.60	3.31	6.12	0.00–23
N	62.80	10.36	43.9–80.50	65.61	19.99	10.3–92.10

**Table 3 jpm-14-00021-t003:** Mann–Whitney U test applied to contents of biological parameters in the case of patients with PCI and multiples comorbidities.

	Rank Sum	Rank Sum	U	Z	*p*-Value	Valid N Severe Form	Valid N Moderate Form
LDH	391.000	312,000	81.000	2.652	0.007	16.00	21.00
CRP	394.5	308.5	77.5	2.759	0.005	16.00	21.00
Concentration of corticosteroids	385.5	317.5	86.5	2.483	0.011	16.00	21.00

**Table 4 jpm-14-00021-t004:** Descriptive statistics for biological variables based on associated comorbidities in patients with PCI.

	LDH	CRP	Glucose	WBC	N	L
**Biological reference interval (BRI)**	100–225 U/L	0–5 mg/L	70–11 mg/dL	4.0–10.0 × 10^3^/µL	34–69%	20–52%
Cases (n = 10) **Post-COVID-19 infection and comorbidities (asthma)**						
Relative risk moderate form/severe form = 0.36/0.63						
Min	167	0.6	87.9	5.53	43.9	4.9
Max	560	298.2	188.5	22.69	92.1	41.1
Mean	276.64	59.06	114.21	11.15	65.57	23.49
Standard deviation	129.37	93.63	28.11	6.10	16.08	11.64
Results ˃ BRI *	6	8	3	4	4	0
Results < BRI	0	0	0	0	0	2
Cases (n = 15) **Post-COVID-19 infection and comorbidities (asthma, hypertension)**						
Relative risk moderate form/severe form = 0.41/0.53						
Min	133	0.6	80.9	5.14	51.5	10.9
Max	478.5	66.4	135.5	12.66	80.5	40.7
Mean	265.85	18.12	108.19	8.99	67.98	21.39
Standard deviation	113.68	19.52	17.10	2.18	10.25	8.63
Results ˃ BRI	8	10	8	7	9	0
Results < BRI	0	0	0	0	0	8
Cases (n = 5) **Post-COVID-19 infection and comorbidities (asthma, hypertension, diabetes mellitus)**						
Relative risk moderate form/severe form = 0.84/1.13						
Min.	171.1	2.1	96.04	4.92	47	19.1
Max.	303.7	38.4	203.2	8.96	72.8	28.9
Mean	228.02	12.08	128.7	6.93	58.46	25.16
Standard deviation	51.49	16.30	48.41	1.73	10.72	5.05
Results ˃ BRI	3	2	2	0	1	0
Results < BRI	0	0	0	0	0	1
Cases (n = 3) **Post-COVID-19 infection and comorbidities (asthma, hypertension, obesity)**						
Relative risk moderate form/severe form = 0.91/1.15						
Cases	3					
Min.	165.1	0.7	84.6	6.36	56	3.5
Max.	255.3	125.9	218.1	16.77	91.7	24
Mean	211.13	43.66	139.7	11.82	71.2	16.93
Standard deviation	45.12	71.24	69.73	5.22	18.4	11.63
Results ˃ BRI	1	1	2	2	1	1
Results < BRI	0	0	0	0	0	2
Cases (n = 4) **Post-COVID-19 infection and comorbidities (asthma, hypertension, diabetes mellitus, obesity)**						
Relative risk moderate form/severe form = 0.98/1.52						
Min.	123	0.6	101.6	5.9	50.3	16.2
Max.	324	21.3	205.7	13.2	77.9	39
Mean	211.2	10.42	140.97	9.95	62.25	28.05
Standard deviation	100.54	10.17	44.94	3.79	11.94	9.61
Results > BRI	2	2	3	2	1	0
Results < BRI	0	0	0	0	0	1

* BRI = biological reference interval.

**Table 5 jpm-14-00021-t005:** Summary of discriminant function analysis for grouping patients with AB, patients with AB, H, DM, patients with AB, H, and patients with AB, H, OB.

Variable	Wilks’ Lambda	Partial Lambda	F(4.28)	*p*	R^2^
ALT	0.309	0.608	4.502	0.006	0.347
Age	0.340	0.552	5.659	0.001	0.258
RBC	0.258	0.729	2.599	0.057	0.33
Urea	0.239	0.788	1.879	0.141	0.284
CRP	0.233	0.807	1.668	0.185	0.226

ALT = alanine aminotransferase; CRP = C-reactive protein; RBC = red blood cell count.

## Data Availability

No new data were created or analyzed in this study. Data are contained within the article.

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
