# Peer review of "Assessment of Some Risk Factors and Biological Predictors in the Post COVID-19 Syndrome in Asthmatic Patients"

_jpm, 2023, doi:10.3390/jpm14010021_

Round 1
Reviewer 1 Report
Comments and Suggestions for Authors
The manuscript “Assessment of some clinical and biological predictors in the post COVID-19 syndrome in asthmatic patients” deals with the determination of the evaluation the epidemiological and clinical risk factors associated with developing post-COVID-19 infection (PCI) among patients with asthma.
The topic discussed is very important in the treatment PCI among patients with asthma. Detected markers could therefore provide a useful screening protocol for predicting comorbidities and will allow to implement a personalized approach to patients with asthma.
I would like to make a few comments:
1) The title does not reflect the main idea of the investigation; perhaps it is better to change it.
2) Line 103: Add please the word “bronchial” if the abbreviation BA is used.
3) Please indicate the race of the patients in the Materials and Methods.
4) Indicate in the captions to the figures and tables the explanation of all abbreviations.
5) In the Discussion please compare the result of eosinophil number obtained in your investigation and the results of the article (898 patients), where the eosinophil count of COVID-19-infected patients with asthma was more higher:
Rawy AM, Sadek MS, Mogahed MM, Khamis A, Allam AH. Relationship between bronchial asthma and COVID-19 infection in adults: clinical and laboratory assessment. Egypt J Bronchol. 2023;17(1):12. doi: 10.1186/s43168-023-00183-9.
6) In the Discussion, it is necessary to mention approaches aimed to determine the markers of phenotype, severity and associated pathologies from other researchers. Cite please the articles in which the description of additional COVID-19 severity markers in asthma patients is described:
M. Babul Islam, Utpala Nanda Chowdhury, Md. Asif Nashiry, Mohammad Ali Moni,
Severity of COVID-19 patients with coexistence of asthma and vitamin D deficiency,
Informatics in Medicine Unlocked, Volume 34, 2022,
https://doi.org/10.1016/j.imu.2022.101116.
(https://www.sciencedirect.com/science/article/pii/S2352914822002532)
Azam Jahangirimehr, Elham Abdolahi Shahvali, Seyed Masoud Rezaeijo, Azam Khalighi, Azam Honarmandpour, Fateme Honarmandpour, Mostafa Labibzadeh, Nasrin Bahmanyari, Sahel Heydarheydari. Machine learning approach for automated predicting of COVID-19 severity based on clinical and paraclinical characteristics: Serum levels of zinc, calcium, and vitamin D,
Clinical Nutrition ESPEN, Volume 51, 2022, Pages 404-411,
https://doi.org/10.1016/j.clnesp.2022.07.011.
(https://www.sciencedirect.com/science/article/pii/S2405457722004211)
7) The limitations of the study should be mentioned: the small number of patients (37patients) and the absence of comparison groups (moderate COVID without asthma; sever COVID without asthma; COVID non-infected with asthma).
Reviewer 2 Report
Comments and Suggestions for Authors
Well written article but the results have limited originality. Many studies have confirmed that asthma does not represent a risk factor for developing severe Covid-19 pneumonia, unlike other comorbidities also described in this paper (arterial hypertension, diabetes, obesity, etc.). Higher CRP and LDH values are also known to be correlated with more severe forms or post-covid infections. It would be appropriate for the authors to explain more clearly what the original insights of this study are and what the results can add to the existing literature in order to improve clinical practice.
Round 2
Reviewer 2 Report
Comments and Suggestions for Authors
The authors have sufficiently addressed the queries and improved the manuscript.